METHODS AND RESOURCES

# qByte: An open-source isothermal fluorimeter for democratizing analysis of nucleic acids, proteins and cells

Francisco J. Quero[1,2]*, Guy Aidelberg[1], Hortense Vielfaure[1], Yann Huon de Kermadec[1], Severine Cazaux[3,4], Amir Pandi[1], Ana Pascual-Garrigos[2], Anibal Arce[3,4], Samuel Sakyi[5], Urs Gaudenz[6], Fernan Federici[3,4], Jennifer C. Molloy[2]*, Ariel B. Lindner [1]*

1 Université Paris Cité, INSERM U1284, Center for Research and Interdisciplinarity, Paris, France, 2 Department of Chemical Engineering and Biotechnology, University of Cambridge, Cambridge, United Kingdom, 3 ANID - Millennium Science Initiative Program - Millennium Institute for Integrative Biology (iBio), Santiago, Chile, 4 Institute for Biological and Medical Engineering, Schools of Engineering, Medicine and Biological Sciences, Pontificia Universidad Católica de Chile, Santiago, Chile, 5 Department of Molecular Medicine, Schoolof Medicine and Dentistry, College of HealthSciences, KNUST, Kumasi, Ghana, 6 GaudiLabs, Lucern, Switzerland

* fjq21@cam.ac.uk (FJQ); jcm80@cam.ac.uk (JCM); ariel.lindner@inserm.fr (ABL)

**Data availability statement:** All relevant data are within its Supporting information files.

## Abstract

Access to affordable and reliable scientific instrumentation remains a significant barrier to the democratization of healthcare and scientific research. In the field of biotechnology, in particular, the complexity, cost, and infrastructure requirements of many instruments continue to limit their accessibility, especially in resource-limited environments. Despite the recent increase in the development of open-source tools, driven by advances in digital fabrication and electronic prototyping, few of these projects have reached large-scale implementation or validation in real-world settings. Here, we present qByte, an open-source, 8-tube isothermal fluorimeter designed to overcome these barriers by offering a cost-effective ($60) yet production-ready solution. qByte leverages standard digital manufacturing and Printed Circuit Board (PCB) assembly techniques and is designed to be portable, making it ideal for both laboratory and field use. The device has been benchmarked against commercial real-time thermocyclers and spectrophotometers, showing comparable results across four key applications: nucleic acid amplification and detection, including the on-site diagnosis of human parasites in Ghana, analysis of protein activity and stability, genetic construct characterization, and bacterial viability tests. Taken together, our results proved qByte as flexible and reliable equipment for a variety of biological tests and applications, while its affordability and open-source design simplify further development and allow adaptation to the needs of future users.

## Introduction

The availability of affordable and reliable scientific instrumentation is a key factor in driving scientific research, especially in low resource settings. Despite significant progress made in recent years, there still exist notable challenges that need to be addressed on the way to

Custom code and all associated resources are available in the gitlab repository of the project at https://gitlab.com/open-bioeconomy-lab/diagnostics-hardware/rt-lamp-device/-/tree/master/qByte?ref_type=heads and also via Zenodo at 10.5281/zenodo.15337451.

**Funding:** Funding for this work was provided by multiple sources: The Bettencourt Schueller Foundation through the Learning Planet Institute (https://www.learningplanetinstitute.org) support to ABL and FJQ, the French Foreign Ministry NYANSAPO grant (https://www.diplomatie.gouv.fr/fr/dossiers-pays/ghana/relations-bilaterales/) support to SS and ABL, Trinity College External Research Studentship (https://www.trin.cam.ac.uk) support to APG, Shuttleworth Foundation Fellowship (https://www.shuttleworthfoundation.org) and the Isaac Newton Trust Strategic Research Award (https://www.newtontrust.cam.ac.uk) support to JCM, the Gathering for Open Science Hardware and the Alfred P. Sloan Foundation (CDF-103/CDF-201, https://sloan.org) support to ABL and FJQ, Chilean National Agency for Research and Development (ANID) through the ANID Millennium Science Initiative Program (ICN17 022, https://www.iniciativamilenio.cl/en/home_en/) support to FF, and the Fondo de Desarrollo Científico y Tecnológico (FONDECYT Regular 1211218 & FONDECYT Regular 1241452) support to FF. The funders had no role in study design, data collection and analysis, decision to publish, or preparation of the manuscript.

**Competing interests:** The authors have declared that no competing interests exist.

democratize science [1]. Price represents one major bottleneck, but others include the high dependence on specialized infrastructure and trained personnel [2], as well as the variability of custom-built equipment and its calibration across different laboratories [3–5]. In the field of biotechnology, this issue is especially pronounced. The required instrumentation is often complex, non-portable, and costly, thus restricting the majority of protocols to well-funded laboratories and facilities that are largely inaccessible to Low and Middle-Income Countries (LMICs).

In this context, the recent emergence of sustainable open-source and low-cost product models has been a major revolution. Projects such as Arduino have laid the foundations for a growing culture of Open Science Hardware (OSH), bringing together large communities of researchers and makers who develop, document, and openly share their designs [6]. The open-source alternatives to traditional proprietary machines range from spectrophotometers [7] to optomechanical platforms [8] and automated 3D printed microscopes [9]. While these projects present a valuable repository of solutions to build upon, only a few projects [9–11] advanced beyond the prototyping phase to full implementation, including validation in real-world scenarios and scaling in production.

Perhaps due to its critical applications in diagnosis and clinical settings, the nucleic acid amplification field stands out from this trend, with several open-source, production-ready kits successfully developed and commercialized [12,13]. However, they still depend on additional instrumentation (such as electrophoresis equipment or transilluminators), looping back to the initial problem of requiring an already established infrastructure.

Building on the premise of developing standalone, production-ready, and affordable scientific equipment, we identified isothermal fluorimeters as versatile devices that can be employed in a wide range of biotechnological applications. Their capability to maintain a tunable constant incubation temperature while simultaneously detecting fluorescence in real-time can support a broad range of applications, from enabling standalone nucleic acid tests for decentralize healthcare in remote areas [14,15] to supporting enzyme producers assess the activity of different batches of locally produced polymerases [16,17].

Current commercial isothermal fluorimeters are priced starting at $3,000 and are often designed for laboratory setups, making them unsuitable for field work. Although there are more affordable open-source solutions, they still have production costs of several hundred USD, their manufacturing is difficult to scale up [18], they have not been validated in real-world scenarios, or lack benchmarking against state-of-the-art solutions [19,20].

Here, we present the qByte, an open-source 8-tube heating fluorimeter designed and validated for a broad range of biotechnological applications. Our device is fully open-source, with all design files available on the project's GitLab under the CERN-OHL-P-2.0 license. qByte is based on digital manufacturing and standard Printed Circuit Boards (PCB) assembly to enable a simple, decentralized production. The device has been benchmarked against commercial instruments, including real-time thermocyclers and isothermal spectrophotometers, showing comparable results while offering lower production costs and greater accessibility. qByte has also been validated in various real-world scenarios, demonstrating its portability and reliability in environmental detection of human pathogens, characterizing enzyme activities, and studying bacterial viability.

The selection of applications was made to highlight the versatility of qByte, spanning different fields—from healthcare-related diagnostics to the democratization of synthetic and systems biology—showcasing its potential beyond a single type of assay. With a production cost of just $60 per unit, we believe that qByte represents a competitive solution for researchers, enzyme producers or healthcare professionals in diverse settings.

## Results

### Design principles

We targeted the qByte to be an affordable, easy-to-use, easy-to-scale, open-source fluorimeter that could be ordered and assembled in a simple and decentralized manner, while assuring reproducibility. To achieve this, we established clear design goals and considered a set of technical requirements (Fig 1A):

1. **Compact and lightweight** design, facilitating transport and portability with a weight below 200 grams and dimensions of 100x90x40mm.
2. **8-tube strips** for multiple reactions in parallel.
3. Configurable **power supply options**, **9V or 12V at 2.5A**, using **USB-C** standard. **External battery** support, making the whole system **portable**.
4. **WiFi connectivity** for remote control and data sharing.
5. Sensitivity range able to measure DNA amplification kinetics, cell-free protein synthesis or cell viability tests.
6. **Reproducibility across tubes and devices**, in terms of thermal uniformity and fluorescence kinetics detection.
7. **Standalone software**, with no installation required, easing the control and monitoring of experiments on-the-go by any device in any place.
8. Temperature ranges from **environmental temperature up to 95 °C**, as well as a **heated lid** to avoid condensation in the upper part of the tube.
9. Adapted for **mass production**, with standardized components and assembly processes that minimize production costs and ensure consistent quality across multiple devices.

To test and refine the design features of our device, we followed an iterative prototyping process, initially focusing on each module separately. This approach led to the implementation of the heated lid, enhancements to the optical reading module, and optimization of the 3D printed parts. Detailed information on the lessons learned during this prototyping phase can be found in Data A of S1 File.

### Hardware subsystems

The qByte hardware is divided into two submodules: the structural and the electronic parts. The entire bill of materials (BOM) can be found in the project repository and Data B in S1 File, while the instructions to assemble the system are detailed in the Methods.

**Structural components.** The structural components are divided in three parts: a 3D printed case, a 3D printed aluminum tube holder and a filter holder. The 3D printed case insulates the equipment from the environment and secures the optical and thermal components, that surrounds the reaction tubes held inside the 3D printed aluminum holder. The tube holder presents openings on the sides and on the bottom, respectively allowing the light-emitting diode (LED) light to reach the tube chamber and the resulting fluorescence signal to reach the phototiode. The filter holder is an intermediate 3D-printed piece holding a plastic filter, which allows only the fluorescence signal wavelengths through. The holder lies between the metal part and the measurement circuit and, given its contact with the heat block, it must be printed in a temperature-resistant material, such as polyethylene terephthalate glycol (PETG) (Fig 1A).

**Electronic components.** The qByte electronics are structured in four parts: (1) a temperature control subsystem (heated lid and tube heat block), (2) an optical measurement system

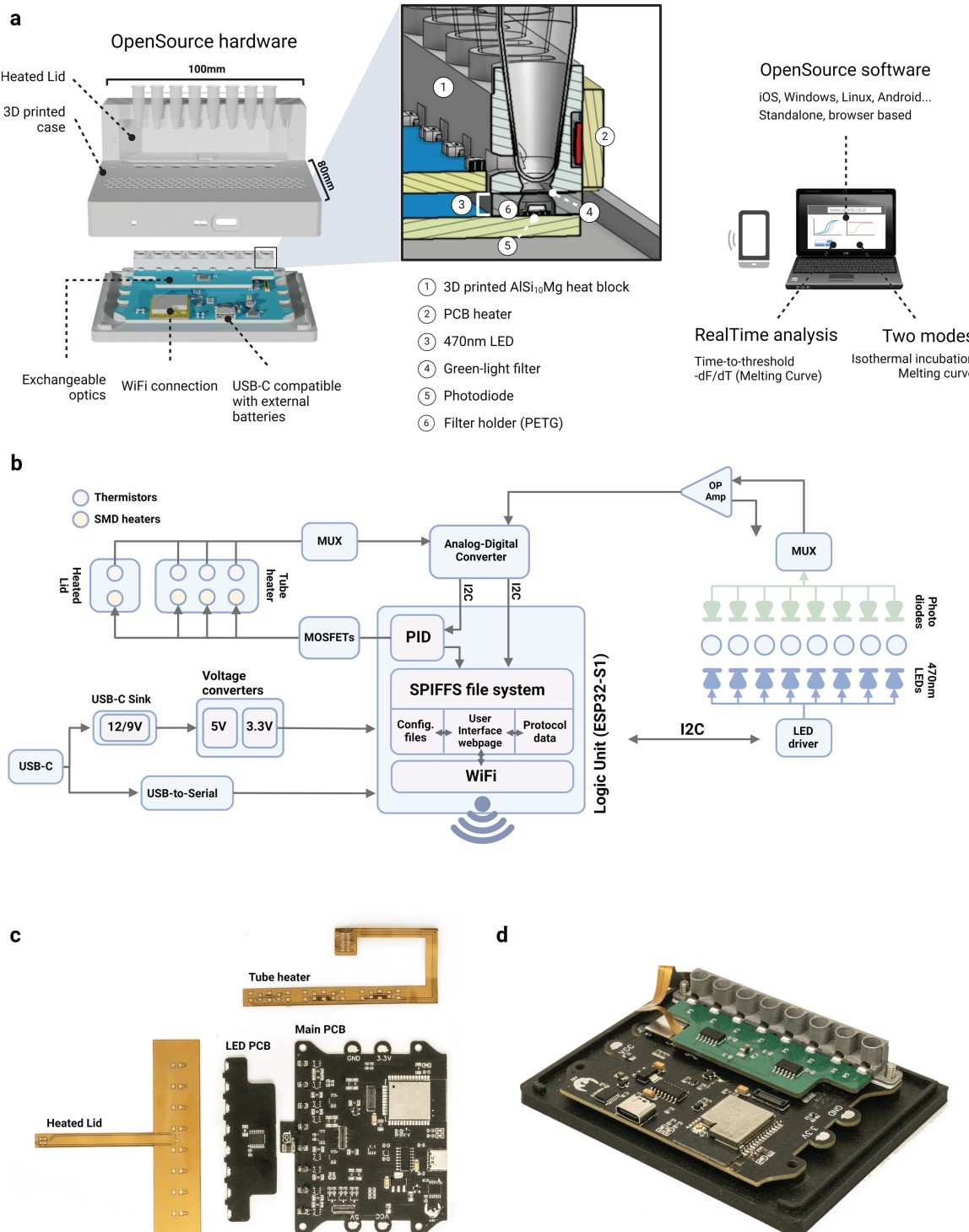

**Fig 1. qByte architecture.** Design principles and general architecture of the qByte. A: Diagram of system and components. B: Hardware architecture, showing each electrical component and its connectivity. C: Top view of the electronic boards. D: Electronics assembled onto the bottom piece of the case.

driving the LED emission and fluorescence signal readout, (3) a logic control system coordinating the sensors and actuators, and (4) a power stage (USB-C standard), that supplies the boards with 3 main voltages and handles the serial communications (Fig 1B).

Temperature control: The heated lid and heat-block heater are built on flexible printed circuit boards (PCBs) containing 100Ω SMD resistors (1.5 Watts) that work as the primary heating elements. They both have Negative Temperature Coefficient (NTC) thermistors that measure the temperature in real-time (Fig 1C). The heated lid has a single heating zone, with one sensor in the middle and eight resistors (one per tube). The tube holder heater is divided into three sectors, each with two sensors and three heaters. This enables the three temperatures to be controlled separately, compensating for possible differences between the edges of the block and the center. A 0.2mm aluminum stiffener on the bottom side of the PCB ensures uniform distribution of the heat and acts as a structural support in the connector and corner areas. The main microcontroller drives each heating segment across both PCBs using a software-based Proportional–Integral–Derivative (PID) algorithm, controlling an individual Metal–Oxide–Semiconductor Field-Effect Transistor (MOSFET). All the control components are located on the main board, with only the sensors and heating resistors externalized to the flexible PCBs.

Optical measurement: The optical subsystem of the device consists of three components: the light-emitting printed circuit board (PCB), the optical filter, and the circuit for measuring and amplifying the fluorescent signal.

1. Light-Emitting PCB: LEDs mounted on this board light the reaction chamber (470 nm wavelength) while the brightness and burn-out detection are controlled by the LED driver (TLC59108, Texas Instruments). The PCB is interchangeable and the connector exposes the 5V lane and I2C bus, allowing users to design their own modules with different wavelengths. The orthogonal placement of the LED in relation to the photodiode decreases background noise. This layout enables a volumetric working range as low as 10 μL, with an optimal volume of 20 μL or more. Below this threshold, the light hits in the liquid-air interface, which diffracts and prevents coherent and precise measurements.
2. Optical Filter: A plastic sheet filter (JAS Green, LEE Filters) filters the light in the 490-600 nm range, filtering out the LED background. This filter can be easily replaced, allowing users to adapt the system to any other excitation or signal wavelengths.
3. Analog sensor Circuit: The fluorescent signal is detected by one photodiode per tube (TEMD6200FX01, Vishay), and the generated currents are switched by a multiplexer into a high-precision operational amplifier (LMP7717, Texas Instruments) that converts them into voltage. The voltage signal is then measured by a 12-bit Analog-Digital Converter (ADC) (TLA2024, Texas Instruments) and transmitted to the main processor through the I2C bus.

Logic control: The microcontroller (ESP32-WROOM-32, Espressif Systems) is the central processing unit in the logic control subsystem, handling data processing and storage, user interface (UI) management, and actuator control.

1. Data processing and storage: Sensor data from both optical and thermal systems are collected through the ADC via the I2C bus and stored in the ESP32 flash memory.
2. UI management: The microcontroller also acts as a server, providing a UI in the form of a web page accessible via computers or smartphones, through the local network.

3.  Actuator control: The microcontroller controls the LEDs, via I2C communication with their driver, and the heaters, feeding a PID computed Pulse Width Modulated (PWM) signal into the control MOSFETs.

## Software subsystems

The software is built on two components: an embedded system programmed in Arduino language, and an UI programmed in HTML5, CSS and JavaScript. The frontend is stored on the device's flash memory and it is served once the device is accessed through its local IP or mDNS address (Figure 1b).

The control website has four tabs: experiment design, run, analysis, and calibration—the latter being used exclusively for calibration prior to product use.

Users can customize their experimental layout in the design tab. Then, the run tab allows the user to choose between isothermal incubation and melting curve, with adjustable parameters such as temperature and sampling rate. In melting mode, users can set the initial and final temperatures, the temperature increase per step, and the duration of each step. Finally, the analysis tab is divided into 'time-to-threshold' and 'melting curve' mode. With the first, users can select data ranges, normalization methods and fluorescence thresholds: the software then generates a results table. In the second, the software displays the computed graphs, and allows users to select the window size to compute the derivative.

Additional details about the embedded system and the UI design can be found in Data D and E of S1 File.

## Application 1: Nucleic acid tests

Nucleic acid amplification tests (NAATs) are widely used in healthcare, agriculture, and environmental monitoring. Polymerase chain reaction (PCR) [21,22], and specifically its real-time quantitative variation (qPCR), has become the gold standard in molecular analysis of DNA and RNA [23], due to its capacity to amplify from as few as 1 to 10 molecules per reaction and quantifying target concentrations.

Isothermal Nucleic Acid Tests (iNAAT), and specifically loop-mediated isothermal amplification (LAMP) [24,25], are increasingly being seen as a robust alternative to PCR. Their ability to amplify at a constant temperature and overcome the effects of amplification inhibitors streamline the process and simplify the required equipment [26], making the technique particularly suitable for deployment directly on the field or resource-limited settings.

In order to compete with qPCR, LAMP is required to perform real-time measurements of the amplification signal, which not only allows to discern potential false positives but also to estimate the approximate DNA concentration in the sample [27]. Furthermore, real-time measurements help correlate the impact of reaction conditions on kinetics, thereby enabling local reagent producers and researchers to fine-tune their own recipes for their specific cases [28].

However, the existing literature on real-time LAMP hardware still presents challenges in scaling production [18], reliance on colorimetric sequence-unspecific amplification methods that makes the system prone to false positives [27,29], and difficulties maintaining heat uniformity across reactions [19]. In this regard, the qByte aims to offer a resilient open-source option, combining sensitivity, manufacturability, portability and replicability with cost-effectiveness.

Our results show that, when compared to commercial qPCR machines, the qByte performs at a similar level (Fig 2A, 2B). It can effectively distinguish between three concentrations of

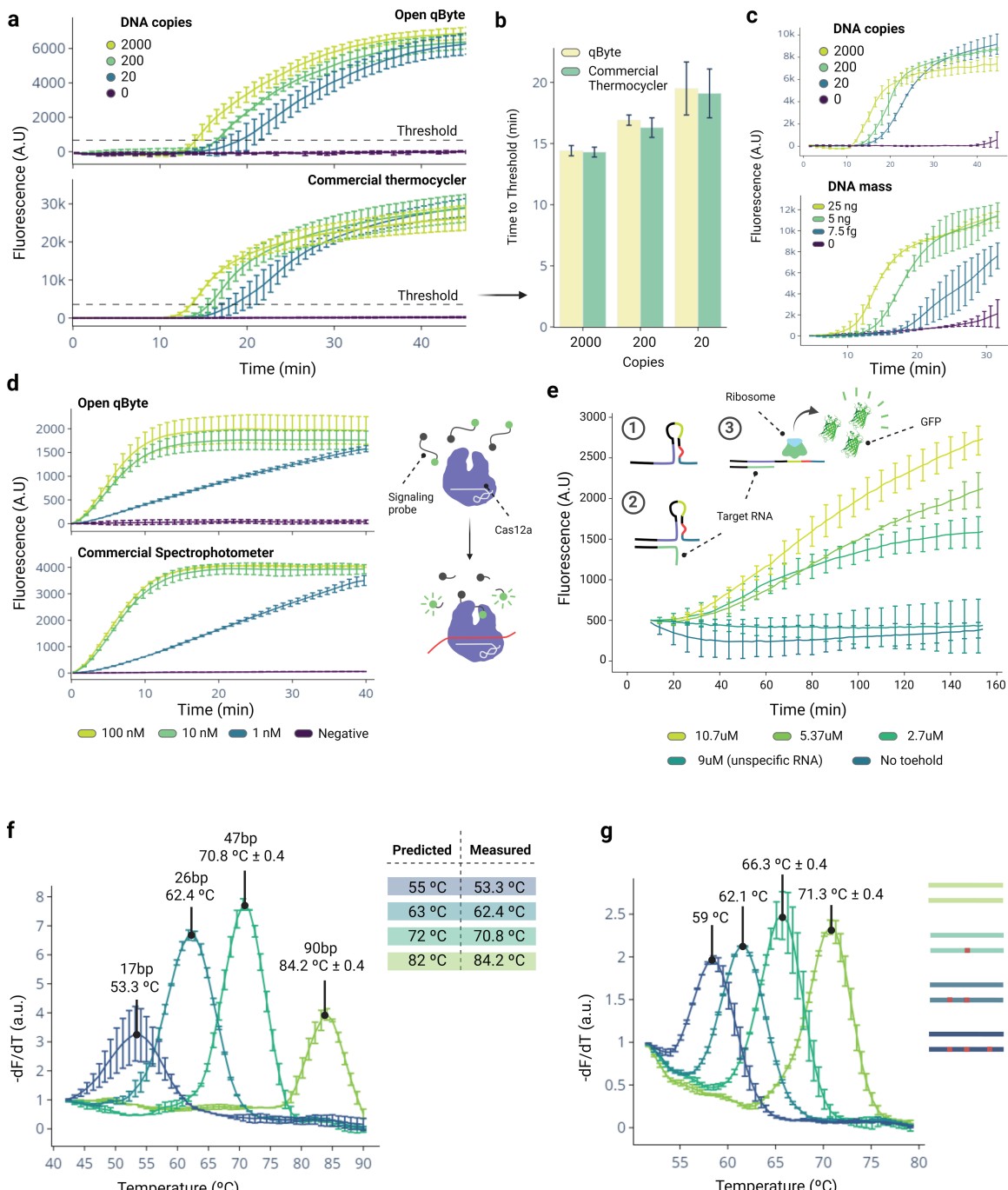

**Fig 2. Results of qByte for nucleic acid tests.** A: Benchmark of the qByte and a commercial thermocycler for performing isothermal amplifications using GMO detective reactions [30]. B: Time-to-threshold comparison. C: Further validation using CoronaDetective reactions [31] (top) and previously developed LAMP reactions for *Schistosoma mansoni* detection, performed on-site at a healthcare facility in Ghana (bottom). D-E: Benchmark of the qByte for non-amplification detection using Cas12a with synthetic Salmonella enterica serovar Typhi (*S. typhi*) DNA (D) and Zika virus RNA fragment detection via activation of an RNA toehold (E). F-G: Analysis of the impact on the melting curve of different oligo sizes (F) and the effect of increasing mismatch mutations (G). The raw data supporting all figures can be found in S2 File.

transgenic samples in agricultural crops [30], achieves this with a time-to-threshold comparable to the commercial solution, and successfully detected SARS-CoV-2 RNA in Viral Transport Medium (VTM). Lastly, to test the system in a real-world scenario, we employed the qByte at a local facility in Ghana, discriminating between three different concentrations of *Schistosoma mansoni* DNA from purified urine samples (Fig 2C).

We also evaluated the performance of qByte in non-amplification-based DNA assays by testing: 1) the detection of a synthetic Salmonella enterica serovar Typhi (*S. typhi*) DNA sequence using the LbaCas12a enzyme [15] (Fig 2D), and 2) the detection of a Zika virus RNA fragment by triggering the activation of an RNA toehold switch that results in the production of Green Fluorescent Protein (GFP) in cell lysates [32] (Fig 2E). In both cases, qByte achieved performance comparable to a commercially available solution, a 96-well plate spectrophotometer.

Finally, we tested qByte for melting curve analysis, which aids in identifying false positives [33] or detecting single nucleotide polymorphisms (SNPs) to distinguish between different genome variants [34] by studying the DNA melting temperature in a sample. Using the qByte melting curve mode, we 1) compared the melting points of synthetic oligonucleotides of varying lengths and 2) assessed the impact of an increasing number of mutations—intended to simulate single nucleotide polymorphisms (SNPs)—on the melting point between a primer and its target (Fig 2F, 2G). The results illustrate qByte's ability to differentiate melting points of a single SNP and the variations in oligo size. The sequences used in this study were derived from a SARS-CoV-2 diagnosis amplicon [35], highlighting the qByte potential for identifying either false positives or monitoring the emergence of variants through SNP analysis.

## Application 2: Protein activity and stability

For enzyme producers in remote areas, having an affordable and portable tool to reliably characterize enzyme activities across different batches would be extremely useful, especially in diagnostics, where adjusting the enzyme activity is a key parameter for balancing false positives and negatives. Similarly, being able to study enzyme stability in different reaction conditions would help optimize mixtures to reduce dependency on cold chain transportation and storage, which may not always be available in field settings.

We tested the qByte's ability to measure protein activity and stability using an in-house expressed and purified Bst-LF, which is the originally used enzyme in LAMP reactions [24, 28]. We evaluated its performance through two independent assays. First, we studied the activity of Bst-LF by comparing varying concentrations of the in-house produced enzyme versus the standard concentration of a commercial enzyme. By comparing the time-to-threshold results, a local producer can calibrate how much of the produced batch has to add to a final reaction to achieve results similar to those obtained with a commercial enzyme. Additionally, by measuring the activity per gram of enzyme produced, the producer can approximate the specific activity and assess the efficiency of their production process (Fig 3A).

We then demonstrated the qByte's ability to measure protein stability using the melting curve mode to analyze protein denaturation. This would enable local enzyme producers to identify potential candidates that could stabilize enzymes at room temperature, reducing the need for cold storage. To test this, we used ammonium chloride ($NH_4Cl$), reported in the literature as a potential viral protein stabilizer [36]. We performed a thermal shift assay and measured the change in fluorescence, which directly correlates with protein denaturation, at four different $NH_4Cl$ concentrations. Contrary to what has been shown for viral proteins, we found that $NH_4Cl$ negatively affects the stability of the specific in-house Bst-LF (Fig 3B).

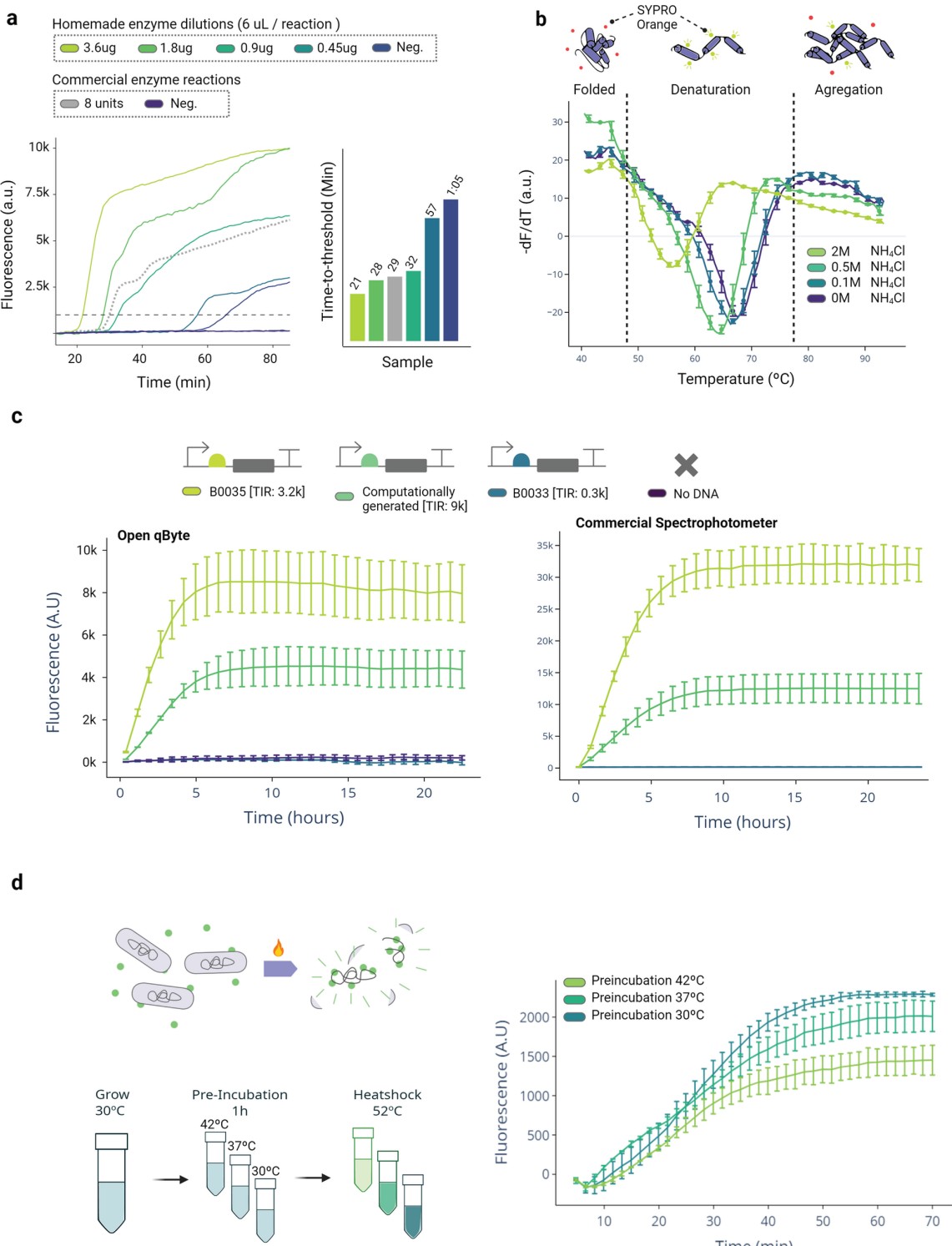

**Fig 3. Validation of qByte for a broad spectrum of applications.** A: Characterization of the enzymatic activity of an in-house produced batch of Bst-LF by comparing it to the standard concentration of a commercial enzyme. B: Validation of the qByte for studying the thermal stability of an enzyme by analyzing the impact of ammonium chloride on the structural stability of Bst-LF using the melting curve mode. C: Benchmarking the qByte against a commercial spectrometer for characterizing the translational efficiency of three constructs for GFP production in the cell-free transcription-translation (TX-TL) system. TIR: translation initiation rate. D: Application of the qByte for studying bacterial viability, specifically measuring the resilience of three bacterial populations pre-incubated at different temperatures to withstand a heat-shock. The raw data supporting all figures can be found in S2 File.

## Application 3: Genetic circuit characterization

The ability to rapidly design, test and benchmark the efficiency and performance of genetic constructs is a handy tool in applications as *in silico* genetic modeling [37]. With the rise of cell-free transcription-translation (TX-TL) systems, this process has been significantly streamlined, bypassing time-consuming transformation protocols, and allowing different constructs to be expressed simply by adding plasmid or linear DNA into a solution [38]. Although the protocols have become more accessible by the recent development of low-cost solutions for cell lysate production [32], the real-time assessment of construct efficiencies still relies on costly laboratory equipment.

Aiming to demonstrate qByte's versatility beyond diagnostic applications, we explored its use in combination with TX-TL reactions to perform the characterization of different protein expression constructs. We characterized the translational efficiency of three distinct ribosome binding sites (RBS): two that have been previously studied [39], and a novel one designed *in silico* using state-of-the-art computational algorithm [40]. Despite a mild increase in noise, the qByte shows a kinetics comparable to a commercial spectrophotometer (Fig 3C), proving the potential of the device to be employed to benchmark different constructs.

## Application 4: Bacterial viability

Finally, we assessed the qByte performance in conducting cell viability tests in bacteria. We used the membrane-impermeable DNA intercalating dye SYTOX Green-which generates green fluorescence only when the cell membrane is compromised-as a marker of bacterial viability.

Aiming to replicate previous results showing how pre-conditioned bacteria grown in warmer environments overcome an adaptive heat-shock resistance [41,42], we investigated the resilience of bacterial populations pre-incubated at different temperatures to a sudden 52 °C heat shock. Our data is in agreement with the literature, showing that the bacteria pre-incubated at higher temperatures have lower mortality rates during the heat-shock (Fig 3D).

In this line, a promising area for future testing is antibiotic susceptibility assays, highly relevant to real-world applications, especially in LMICs. Although we have managed to obtain preliminary results of the effect of various bacteriolytic antibiotics on bacterial death rates, the lack of a second OD or fluorescence channel to normalize the signal currently restricts its practical application.

## Discussion

In this study, we introduced qByte, an 8-tube isothermal fluorimeter specifically designed to provide a cost-effective, portable, and reliable alternative to the equivalent commercial instrumentation currently on the market. With a production cost as low as $60 per unit, qByte is based on an open-source design that leverages digital manufacturing and standard PCB assembly easing its production in small numbers yet allowing it to scale when needed. The design does not require individual calibration for each produced unit; the software is standalone and operates on any web browser; and the system is lightweight and can be powered with an off-the-shelf external battery. These features make qByte easy to produce and use on the field, making it accessible to a broad spectrum of users, from researchers in well-equipped labs to enzyme producers in resource-limited environments. The device was specifically designed with the latter settings in mind, aiming to provide a versatile, low-cost solution that supports a wide range of experiments and applications.

We therefore tested qByte by benchmarking against commercial real-time thermocyclers and spectrophotometers across four applications. Our main focus was on nucleic acid detection, because of its wide use in diagnostics and environmental sampling; there, qByte effectively detected nucleic acids through isothermal amplification and non-amplification methods, also supporting melting curve analysis of amplified sequences to check for non-specific amplicons (Fig 2). Furthermore, in our fieldwork in Ghana qByte demonstrated its ability to take nucleic acid-based diagnostics out of the laboratory, allowing for the on-site detection and analysis of human parasite samples (Fig 2C). We then moved to the study of protein activity and stability, where qByte can help local enzyme producers to characterize enzyme batch activities and optimal storage conditions (e.g., buffers, enzymes variants) (Fig 3A, 3B). As a third application, we tested the reliability of qByte to work on genetic constructs characterization on cell-free systems, which can potentially be employed as a simple and affordable alternative to commercial spectrophotometers in determining the construct expression potential (Fig 3C). Finally, as a proof of concept in the context of bacteria viability study, we used qByte to detect heat-shock resistance adaptation in bacteria as described in the literature [42] (Fig 3D). Based on the positive results obtained, we believe that qByte could be applied to a broader range of bacterial viability tests, with antimicrobial resistance heading the list where the lack of an affordable yet precise point-of-care alternative presents a potential opportunity to apply qByte.

We showcased here four ways in which qByte can be applied, yet those represent only an initial demonstration of its capabilities. As other open-source hardware projects have demonstrated (e.g. Arduino), the full spectrum of potential applications becomes evident only after the device is widely produced and adopted by others. For example, one of the main bottlenecks in nucleic acid amplification is not the amplification itself but the preceding steps—particularly sample preparation. Many challenges arise from difficulties in accessing DNA (e.g., from complex environmental samples such as soil or water) or the presence of inhibitory compounds carried over during DNA extraction, which can affect reaction efficiency. The protocols used in this work, adapted from previous studies [30,31], represent a starting point and could be further refined through iterative improvements and open collaborations to help address these critical limitations in the field. Although previous studies have made progress on this topic [43], its specificity to each application makes it beyond the scope of this paper. Nevertheless, we believe that the widespread availability of qByte would provide researchers with a low-cost platform to optimize detection protocols for specific targets.

Taken together, qByte remains an exceptionally low-cost and versatile tool for real-time assays. Compared to other open-source devices—including Miriam, Snodgrass Device, SMART-LAMP, Buultjens Device, and commercial solutions, as summarized in Table A of S1 File—qByte offers the most affordable alternative that is production-ready and has been extensively validated, demonstrating replicable results across diverse applications requiring precise dynamic temperature control and fluorescence readouts.

A key distinction of this work from previous devices is the extensive validation of temperature uniformity. It is essential in quantitative applications to ensure that variations in reaction outcomes are not mistakenly attributed to differences in sample concentration or composition when, in reality, they are caused by temperature fluctuations. To achieve this, we employed not only the internal system sensors but we also validated them employing melting probes translating the actual reaction temperature into a fluorescent signal (Data C S1 File). To demonstrate the robustness of this approach, we conducted comparative analyses across replicates in every application, confirming reliable consistency between qByte and commercial solutions.

qByte was the result of an iterative process of tests and implementations. Even though such an approach allowed for tackling most of the issues encountered and implementing different ideas along the way, there is still space for further development and use. Specifically, we identified two main possible future implementations; expanding the measurements from kinetics to absolute fluorescence readings and integrating more readout channels.

In the current design, qByte tracks relative fluorescence changes rather than absolute intensities across its eight channels, which sufficiently supports our demonstrated applications that rely on monitoring signal dynamics over time. However, achieving precise absolute comparisons would be valuable for quantifying absolute concentrations of cells, DNA, or proteins using fluorometric assays, but would require more advanced calibration protocols. As partially explored in Supplementary Data (Fig C in S1 File, where we use reference FAM concentrations), these complexities extend beyond a simple intercept or gain correction: minor variations in electrical component performance, metal holder manufacturing, LED and photodiode positioning, and plastic tube properties can significantly affect the intensity of received light, overshadowing absolute signal measurements. One approach to improving calibration without relying on manual fine-tuning is to use previously measured melting probes (see application 1: Nucleic Acid Tests and Data C in S1 File) to align fluorescence peaks across a known temperature range. First, the device would adjust the temperature model so that melting peaks coincide across all samples (x-axis calibration). Once aligned, a fluorescence model would be applied to normalize absolute intensity values that correspond to the same intensity across channels (y-axis calibration). While this method presents an interesting avenue for improving absolute fluorescence quantification over the full working range, we believe that further development and validation are beyond the scope of this paper.

The second potential enhancement is adding a second optical channel, either for fluorescence or optical density, to enable ratiometric or absorbance measurements. This would improve signal normalization [44] and enhance the reliability of bacterial studies by referencing fluorescence signals against total cell density [45,46], which is particularly relevant for applications such as antimicrobial sensitivity testing and, more broadly, systems biology.

As presented here, qByte has already proven to work effectively outside of the lab where it was developed, having been used in different laboratories across Europe, South America, and Africa. The collaborations are still active, and we are currently leveraging this work for efforts at Bahia Exploradores in Patagonia, Chile, to study the presence of invasive species DNA in water samples. With this we hope to showcase how leveraging open-source hardware devices as collaborative tools can accelerate the development and field deployment of biotechnology, potentially having a significant contribution in practical situations. We encourage researchers aiming to take their nucleic acid tests beyond the lab, teachers seeking tools for hands-on biotechnology education, and medical providers in need of low-cost yet reliable environmental surveillance tools, to consider qByte as a potential solution. We hope this work helps them produce, adapt, or simply use this device for their needs.

## Methods

### qByte assembly

The multiple steps required to assemble a qByte are detailed in the assembly guide available on the project's GitLab repository [47]. The repository also includes the models for components 3D printing, PCB design and gerber files.

PLA was used to print all plastic structural parts on a Prusa i3 MK3S+ printer (Prusa Research), with the exception of the filter holder, printed with temperature-resistant PETG filament.

The metal tube holder and the PCBs manufacturing was outsourced (PCBWay), including components sourcing, soldering and assembly. For allowing the user to select a 9V power supply instead of 12V, a PCB jumper on the board must be soldered before the initial power up.

An affordable plastic-film light filter was selected as the optical filter (LEE Filters, Ref. 738 JAS Green) and obtained from a local distributor (La Boutique du Spectacle, France).

After assembly, the device was connected to a computer for programming. The code was uploaded to the device, followed by local WiFi network configuration, as described in the GitLab repository.

## LAMP detection

All LAMP reactions were performed in a total volume of 20 $\mu$L. The optimized reaction mix consisted of a final concentration of 1.4 mM of dNTPs, 1X Isothermal Amplification buffer supplemented with an additional 5 mM of $MgSO_4$, 2 $\mu$M of SYTO9 fluorescent dye (S34854, Thermo Fisher Scientific), and 0.32 U/$\mu$L of Bst 2.0 DNA polymerase (M0537L, New England Biolabs). For SARS-CoV-2 detection, an additional 0.32 U/$\mu$L of RTx retrotranscriptase (M0380L, New England Biolabs) was included. Primers were obtained from Integrated DNA Technologies (IDT). Their sequences, specific sets, concentration, and original references are reported in Table B, S1 File.

The reactions were incubated at a constant temperature of 63 °C with a lid temperature of 95 °C. Real-time fluorescence data were acquired at a rate of 10 s.

## Cas12a detection

Cas12 reactions were performed in a total volume of 20 $\mu$L, with 1x NEBuffer 2.1 (B7202, New England Biolabs), 100 nm LbaCas12a (M0653T, New England Biolabs), 1 $\mu$M FQ reporter, 62.5 nm sty16_B gRNA (Table B in S1 File), and 100 nm, 10 nm, 1 nm, or 0 nm of 60 bp target DNA (Table B in S1 File). Reactions were incubated at a constant temperature of 37 °C. Real-time fluorescence data were acquired at a rate of 30 s.

Reactions were prepared in quadruplicate. Two reactions were analyzed by the qByte with a lid temperature of 65 °C and two were analyzed by a microplate reader (Spark Multimode, Tecan) using a gain of 130x and excitation and emission wavelengths of 496 and 517 nm, respectively.

## Melting curve analysis

All oligonucleotides were obtained from Integrated DNA Technologies (IDT), and their sequences are listed in Table B, S1 File.

**Effect of oligo size on melting point.** Double-stranded oligonucleotides were ordered and resuspended to prepare 20 $\mu$L reactions, each containing a final concentration of 10 $\mu$M oligonucleotide, 500 mM NaCl, and 2 $\mu$M SYTO9. To prevent evaporation, 20 $\mu$L of mineral oil was layered over each reaction.

Melting points were assessed by incubating the reactions in qByte from 45 °C to 90 °C. The temperature step size was set at 0.2 °C at a rate of 10 s, with one fluorescence measurement per step.

**Effect of mutation number on melting point.** Single-stranded oligonucleotides carrying different numbers of mutations were orderdered along with a common complementary strand. The sequence was obtained from a previously published SARS-CoV-2 primer set [25]. Single-stranded primers were annealed following the manufacturer's instructions and column purified (K0721, Thermo Fisher Scientific) to remove any excess in salts from the annealing buffer.

Reactions were prepared as above and incubated from 50 °C to 80 °C, using the temperature step size and reading time previously described.

## Toehold switch activation in cell-free reactions

**Cell lysates.** Cell lysates were prepared following the protocol described in the literature [32,48]. First, the lysates were prepared from a culture of *E. coli* BL21 Gold (DE3) JM1 dLac, grown on 2xYT + P media supplemented with carbenicillin and kanamycin to reach an OD600 of 0.6. The culture was then induced with 1 mM IPTG to express T7 RNA polymerase for 90 min, followed by washing, pelletization, and lysis using a FastPrep-24 bead-beater (116004500, MP Biomedicals). Beads were then filtered, and lysates centrifuged to extract the supernatant. After 60 min of a run-off reaction at 37 °C and 50 min of dialysis at 4 °C with S30B buffer, the extracts were aliquoted and stored at –80 °C.

**Toehold switch and trigger.** The Zika-specific toehold switch coupled with deGFP described in the literature [49] was used. After transformation in *E.coli* Top10 in a 200 mL culture, the plasmid was extracted via PureYield Plasmid Midiprep System (A2492, Promega) and concentrated at 30 nM per reaction. The RNA trigger was prepared by in vitro transcription for 16 h at 37 °C using 500 nM of template. RNA was purified using a commercial kit (R1013, Zymo Research).

**Cell-free reaction.** The lysates were supplemented with amino acids, NTPs, tRNAs, and phosphoenolpyruvate, among other components [32,48]. A Zika-specific toehold switch was employed at a concentration of 30 nm, while the RNA trigger was used at concentrations of 10.7 $\mu$M, 5.37 $\mu$M and 2.7 $\mu$M. Additionally, a different Zika trigger was used for demonstration of specificity at a concentration of 9uM. Finally, a no-template control without any RNA trigger was included.

The reactions were carried out in duplicate, incubated in the qByte at 30°C with the heated lid at 40°C. Real-time fluorescence data were acquired at a rate of 30 s.

## Bst-LF activity assay

**Transformation of *E.coli* BL21 (DE3).** BL21 (DE3) cells were transformed by heat-shock with the Bst-LF pOpen vector (44) and cultured on LB-agar supplemented with ampicillin. A single colony was incubated in LB/ampicillin medium for Bst-LF expression.

**Protein expression.** A single colony was used to inoculate 10 mL of LB/ampicillin media and incubated overnight at 37 °C. The culture was diluted to reach 0.1 OD at 600 nm, and then incubated at 37 °C until an OD of 0.6-0.8. Bst-LF expression was induced by adding 0.5 mM IPTG at 30 °C, and incubated for 2 h. Cells were then centrifuged in 50 mL tubes at 3000 rpm.

**Protein purification.** Bst-LF was extracted using a lysis buffer (0.5 M Tris-HCl pH 8, 2 M Urea, 1% Triton). For a 100 mL pellet of Bst-LF induced culture, 10 mL of lysis buffer was added. The pellet was solubilized by up-down pipetting. The solution was incubated at room temperature for 90 min using a rotating wheel and then centrifuged to remove the debris and intact cells at 14000 rpm for 10 min.

An immobilized metal affinity chromatography (IMAC) was used to purify the Bst-LF protein. The nickel IMAC column (HisPur Ni-NTA 3 mL) was equilibrated with 10 column volumes of buffer A (40 mM Tris HCl pH 8, 300 mM NaCl). The supernatant was passed through the column and contaminants were washed using 2.5 column volumes of buffer B (25 mM Tris-HCl pH8, 100 mM NaCl, 30 mM Imidazole). Bst-LF was eluted with buffer C (25 mM TrisHCl pH 8, 100 mM NaCl, 150 mM Imidazole). The fractions containing the protein were pooled and diluted with buffer D (10 mM Tris-HCl pH 7.1, 50 mM KCl, 1 mM DTT, 0.1 mM EDTA, 50% glycerol). Ultrafiltration via 50 kDa MWCO columns (Vivaspin), followed by centrifugation at low speed (500-1000 rpm), was used to concentrate the protein.

**Activity assay.** The LAMP assays were conducted according to previously described methods, using our in-house synthesized Bst-LF instead of the commercial enzyme. The sample dilutions for testing were prepared by performing a twofold serial dilution of the initial Bst-LF batch in buffer D.

## Thermal shift assay

Thermal shift assays were conducted in a total volume of 25 $\mu$L. The reactions were prepared with a final concentration of 75 $\mu$g/mL of the protein, 5 mM SYPRO Orange fluorescent dye (S5692, Sigma-Aldrich), and ammonium chloride at final concentrations of 0, 0.1, 0.5, and 2M. The assays were layered with 20 $\mu$L of mineral oil on the top to prevent evaporation and condensation on the cap of the tubes. Reactions were run from 40°C to 95°C, with a step of 0.2°C at a rate of 10 s and one measurement per step, using the qByte and a commercial thermocycler (CFX96, BioRad).

## In vitro protein expression using PURExpress

The in vitro synthesis of proteins was performed using the PURExpress In vitro Protein Synthesis kit (E6800, New England Biolabs). Four different sequences, composed of a T7 promoter, one of the three RBS detailed in Table B, S1 File, followed by the sequence of superfolded GFP (sfGFP), were added to the reactions.

Reactions were assembled on ice in a total volume of 25 $\mu$L, containing 10 $\mu$L Solution A, 7.5 $\mu$L Solution B, 0.5 $\mu$L Murine RNase inhibitor, 5 $\mu$L nuclease-free $H_2O$, and 2 $\mu$L DNA template (250 ng). Solutions A and B were thawed on ice and gently mixed by pipetting. Reactions were incubated at 37 °C for 2 to 4 h and then placed on ice.

The reactions were carried out in duplicate, incubated in the qByte at 37 °C with the heated lid at 50 °C. Real-time fluorescence data were acquired every 3 min overnight.

## Bacterial viability assay

Bacteria were initially grown at 30 °C in Falcon tubes until reaching mid-log phase. Following this, bacterial cultures were split into three separate groups and subjected to pre-incubation at either 30 °C, 37 °C, or 42 °C for 1 h

After pre-incubation, all cultures were washed three times in PBS to remove dead bacteria and residual DNA, pelleted, resuspended to an OD of 0.5 in PBS, and transferred into 0.2 mL tubes containing SYTOX Green at a concentration of 5 $\mu$M. Cultures were then subjected to a heat shock at 52 °C.

Real-time fluorescence data were collected using the qByte device to monitor bacterial viability. Fluorescence was measured at 1 min intervals for 70 min.

## Supporting information

**S1 File. Detailed design, calibration data, and genetic sequences used in this work. S1A Data**, Hardware Prototyping Insights. **S1B Data**, Components Summary and Costs. **S1C Data**, Hardware Calibration. **S1D Data**, Embedded System Design. **S1E Data**, User Interface Design. **S1G Data**, DNA sequences.
(PDF)

**S2 File. Raw data from relevant graphs and results.**
(XLSX)

## Acknowledgments

We further wish to thank Flaminia Zane for her help in the manuscript edition and the Maker Lab at the Learning Planet Institute for their technical support.

## Author contributions

**Conceptualization:** Francisco J. Quero, Guy Aidelberg, Urs Gaudenz, Fernan Federici, Jennifer C. Molloy, Ariel B. Lindner.

**Data curation:** Francisco J. Quero.

**Formal analysis:** Francisco J. Quero.

**Funding acquisition:** Francisco J. Quero, Fernan Federici, Jennifer C. Molloy, Ariel B. Lindner.

**Investigation:** Francisco J. Quero, Guy Aidelberg, Hortense Vielfaure, Yann Huon de Kermadec, Severine Cazaux, Amir Pandi, Ana Pascual-Garrigos, Anibal Arce, Urs Gaudenz, Fernan Federici, Ariel B. Lindner.

**Methodology:** Francisco J. Quero, Guy Aidelberg, Urs Gaudenz, Fernan Federici.

**Project administration:** Ariel B. Lindner.

**Resources:** Francisco J. Quero, Samuel Sakyi, Urs Gaudenz, Fernan Federici, Jennifer C. Molloy, Ariel B. Lindner.

**Software:** Francisco J. Quero.

**Supervision:** Guy Aidelberg, Samuel Sakyi, Urs Gaudenz, Fernan Federici, Jennifer C. Molloy, Ariel B. Lindner.

**Validation:** Francisco J. Quero.

**Visualization:** Francisco J. Quero.

**Writing – original draft:** Francisco J. Quero.

**Writing – review & editing:** Francisco J. Quero, Guy Aidelberg, Severine Cazaux, Amir Pandi, Ana Pascual-Garrigos, Anibal Arce, Urs Gaudenz, Fernan Federici, Jennifer C. Molloy, Ariel B. Lindner.

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
