## [Editor Report · Decision Letter 0]

15 Nov 2024

Dear Dr Lindner,

Thank you for submitting your manuscript entitled "qByte: Open-source isothermal fluorimeter for democratizing analysis of nucleic acids, proteins and cells" for consideration as a Methods and Resources Article by PLOS Biology. Please accept my sincere apologies for the delay in getting back to you with feedback this week. 

Your manuscript has now been evaluated by the PLOS Biology editorial staff and I am writing to let you know that we would like to send your submission out for external peer review.

Once your full submission is complete, your paper will undergo a series of checks in preparation for peer review. After your manuscript has passed the checks it will be sent out for review. To provide the metadata for your submission, please Login to Editorial Manager (https://www.editorialmanager.com/pbiology) within two working days, i.e. by Nov 17 2024 11:59PM.

Kind regards,

Richard

Richard Hodge, PhD

rhodge@plos.org

PLOS

---

## [Decision Letter · Decision Letter 1]

30 Jan 2025

Dear Dr Lindner,

Thank you for your patience while your manuscript "qByte: Open-source isothermal fluorimeter for democratizing analysis of nucleic acids, proteins and cells" went through peer-review at PLOS Biology. Apologies for the delay in sending our decision on your submission. Unfortunately we encountered some delays during the review process. Your manuscript has now been evaluated by the PLOS Biology editors, an Academic Editor with relevant expertise, and by several independent reviewers.

In light of the reviews, which you will find at the end of this email, we are pleased to offer you the opportunity to address the comments from the reviewers in a revision that we anticipate should not take you very long. Some of the technical points made by Reviewer 1 will be important to address by inclusion of additional data, but most other points can be addressed by textual revisions changes. We will assess your revised manuscript and your response to the reviewers' comments with our Academic Editor aiming to avoid further rounds of peer-review, although might need to consult with the reviewers, depending on the nature of the revisions.

**IMPORTANT - SUBMITTING YOUR REVISION**

*Resubmission Checklist*

*Published Peer Review*

*PLOS Data Policy*

*Blot and Gel Data Policy*

Sincerely,

Christian

Christian Schnell

Senior Editor

PLOS Biology

cschnell@plos.org 

on behalf of 

Richard Hodge, PhD

Senior Editor

PLOS Biology

rhodge@plos.org

REVIEWS:

Reviewer #1: see pdf

Reviewer #2: The authors have presented a compelling device that can significantly democratize access to biotechnology research. They present application examples with results that show a diversity of successful uses. There are some minor points which might be addressable: a) Lines 312-332 "Specifically, we identified two main possible implementations…." in which the authors describe absolute versus fluorescence change within the samples and so forth. Further in line 324 they describe a second potential improvement. Overall, this paragraph is confusing because the as a reader, I am not sure if you are describing a set of design choices (First vs Second) that you had to take. I am not sure exactly which one you actually took. It left me with more questions about how the device works than answers. Or if you are discussing future considerations. I would restructure that paragraph in a way that it becomes clear and not intermingle theoretical improvements with actual choices. The data speaks for itself and the work is good, this *paragraph* did not increase confidence in your choices. I am sure that was not your intention. 

b) Your selection of a wide variety of protocols and applications was refreshing to see beyond a diagnostic approach. Perhaps consider framing the selection in terms of diversity (given the versatile nature of your device) or another theme. Otherwise the use of TX-TL reactions seems to be slotted in. 

c) Given your focus on field deployability: The deployment of the device in Ghana is very well explained, as is the work in the UK (Cambridge), Switzerland (Gaudi Labs), France ("Universit´e Paris Cit´e, INSERM U1284, Center for Research and Interdisciplinarity, F-75006 Paris, France) and in Chile ("Institute for Biological and Medical Engineering, Schools of Engineering, Medicine and Biological Sciences, Pontificia Universidad Catolica de Chile, Santiago, Chile"). This alone is a powerful demonstration of international technology transfer. On the other hand, Line 336 (and its summary in the abstract) detracts from this claim since you omit results (Line 337) in the paper. It seems to be a distraction. Suggestion: Omit references to field work whose results you are not showing or position it as future work. 

In summary, the authors have assembled a formidable interdisciplinary team resulting in a real-world, affordable devices with remarkable international results and implications for democratized biotechnology. This work should be published as soon as possible.

---

## [Decision Letter · Decision Letter 2]

27 Mar 2025

Dear Dr Lindner,

Thank you for your patience while we considered your revised manuscript "qByte: Open-source isothermal fluorimeter for democratizing analysis of nucleic acids, proteins and cells" for publication as a Methods and Resources Article at PLOS Biology. This revised version of your manuscript has been evaluated by the PLOS Biology editors, the Academic Editor and the original reviewers.

Based on the reviews, I am pleased to say that we are likely to accept this manuscript for publication, provided you satisfactorily address the following data and other policy-related requests that I have provided below (A-E):

(A) We would like to suggest the following minor edit to the title:

“qByte: an open-source isothermal fluorimeter for democratizing analysis of nucleic acids, proteins and cells”

(B) You may be aware of the PLOS Data Policy, which requires that all data be made available without restriction: http://journals.plos.org/plosbiology/s/data-availability. For more information, please also see this editorial: http://dx.doi.org/10.1371/journal.pbio.1001797

-Supplementary files (e.g., excel). Please ensure that all data files are uploaded as 'Supporting Information' and are invariably referred to (in the manuscript, figure legends, and the Description field when uploading your files) using the following format verbatim: S1 Data, S2 Data, etc. Multiple panels of a single or even several figures can be included as multiple sheets in one excel file that is saved using exactly the following convention: S1_Data.xlsx (using an underscore).

-Deposition in a publicly available repository. Please also provide the accession code or a reviewer link so that we may view your data before publication. 

Figure 2A-G, 3A-D, S1C (Figures 2 and 3)

(C) Please also ensure that each of the relevant figure legends in your manuscript include information on *WHERE THE UNDERLYING DATA CAN BE FOUND*, and ensure your supplemental data file/s has a legend.

(D) Please note that we cannot accept sole deposition of code in Github/Gitlab, as this could be changed after publication. However, you can archive this version of your publicly available code to Zenodo. Once you do this, it will generate a DOI number, which you will need to provide in the Data Accessibility Statement (you are welcome to also provide the GitHub access information). See the process for doing this here: https://docs.github.com/en/repositories/archiving-a-github-repository/referencing-and-citing-content

(E) Please ensure that your Data Statement in the submission system accurately describes where your data can be found and is in final format, as it will be published as written there. 

We expect to receive your revised manuscript within two weeks. 

*Published Peer Review History*

*Press*

Best wishes,

Richard

Richard Hodge, PhD

rhodge@plos.org

Reviewer remarks:

Reviewer #1 (Saad Bhamla, signs review): Authors have addressed all my comments

Reviewer #2: Congratulations!

---

## [Editor Report · Decision Letter 3]

6 May 2025

Dear Ariel,

On behalf of my colleagues and the Academic Editor, Tom Misteli, I am pleased to say that we can accept your manuscript for publication, provided you address any remaining formatting and reporting issues. These will be detailed in an email you should receive within 2-3 business days from our colleagues in the journal operations team; no action is required from you until then. Please note that we will not be able to formally accept your manuscript and schedule it for publication until you have completed any requested changes.

In addition, please include a sentence in the figure legend for S1C Figure 3 in the S1 Data file that states where the underlying data can be found (‘The raw data supporting all figures can be found in S2 File’). This can be included during the production process. I have also included the PDF version of the manuscript in the File Inventory as a Supplementary File. This can be removed from the File Inventory if needed during production. 

PRESS

Best wishes, 

Richard

Richard Hodge, PhD

rhodge@plos.org

PLOS
